# Serum IgG titer findings for *Fusobacterium nucleatum* associated with clinical outcome following surgery in patients with esophageal squamous cell carcinoma

Toru Yoshikawa[1], Hiromi Nishi[2]*, Manabu Emi[1], Yoichi Hamai[1], Yuta Ibuki[1], Tomoaki Kurokawa[1], Ryosuke Hirohata[1], Manato Ohsawa[1], Nao Kitasaki[1], Hitoshi Komatsuzawa[3], Hiroyuki Kawaguchi[2], Morihito Okada[1]

1 Department of Surgical Oncology, Hiroshima University, Hiroshima, Japan, 2 Department of General Dentistry, Hiroshima University Hospital, Hiroshima, Japan, 3 Department of Bacteriology, Graduate School of Biomedical and Health Sciences, Hiroshima University, Hiroshima, Japan

* hiyoko@hiroshima-u.ac.jp

## Abstract

*Fusobacterium nucleatum* (*Fn*) is known as an etiological factor related to periodontitis. However, recent reports have shown that it also functions as an oncogenic pathogen and is associated with progression of various cancers as well as poor prognosis of esophageal squamous cell carcinoma (ESCC) patients. The present study was conducted to examine the correlations of serum IgG antibody titer level against *Fn* (IgG-*Fn*) with clinicopathological characteristics, oral conditions, and survival outcomes in 305 patients who underwent an esophagectomy procedure for ESCC. The results revealed that 40.7% of the patients tested positive for IgG-*Fn*, and those in the positive group had significantly lower rates of overall survival (OS) and cancer-specific survival (CSS) as compared to patients who tested negative (OS; $p = 0.01$, CSS; $p = 0.02$). Multivariate analysis showed IgG-*Fn* positivity as an independent predictor of CSS (hazard ratio, 1.96; 95% CI, 1.32–2.90; $p < 0.001$). Moreover, among patients who were diagnosed with stage II-IV and underwent neoadjuvant therapy (NAT), those in the IgG-*Fn*-positive group showed higher pathological progression and greater incidence of inadequate response to NAT as compared to the IgG-*Fn*-negative group (odds ratio 0.48, 95% CI 0.24–0.92; $p = 0.04$). The present findings indicate that IgG-*Fn* can serve as a marker indicating poor tumor response to NAT in advanced ESCC cases.

## Introduction

Esophageal cancer ranks sixth among the leading causes of cancer-related deaths and has been reported as the most prevalent type of carcinoma worldwide [1]. Notably, comparisons of countries have shown that esophageal squamous cell carcinoma

**Data availability statement:** All relevant data are within the manuscript and its Supporting Information files.

**Funding:** HN was supported by the Japan Society for the Promotion of Science (JSPS KAKENHI) under grant numbers 20K10162 and 23K09311 (https://www.jsps.go.jp/english/). The funder had no role in study design, data collection and analysis, decision to publish, or preparation of the manuscript.

**Competing interests:** The authors have declared that no competing interests exist.

**Abbreviations:** CEA: carcinoembryonic antigen; ECOG PS: Eastern Cooperative Oncology Group performance status; EGJ: esophagogastric junction; Fn: *Fusobacterium nucleatum*; IgG: immunoglobulin G; SCC: squamous cell carcinoma-related antigen.

(ESCC) is the predominant histologic type of esophageal cancer in Eastern Asia. Despite advancements in treatments, the prognosis of patients with ESCC remains bleak [2], and there is a pressing need for research to clarify its pathogenesis, and investigations of novel options for diagnosis and treatment.

Periodontal disease is often caused by infection from oral bacteria and may result in tooth loss [3] or an infectious disorder when receiving care for a systemic disease such as cancer. Thus, oral care during the perioperative period is important [4]. In addition, studies have indicated that periodontal disease can affect development and progression of various types of cancers including esophageal cancer, indicating that the human oral microbiome may play an important role when considering treatment approaches [5].

*Fusobacterium nucleatum* (*Fn*), a type of gram bacteria that can be found in both the oral and gastrointestinal tracts in humans, is associated with periodontal disease development [6]. Findings suggesting that *Fn* may play a role as an oncogenic pathogen and have an association with progression of ESCC have been presented [7], while others have shown that *Fn* could be associated with resistance to neoadjuvant chemotherapy in ESCC patients [8].

The serum IgG antibody titer against bacteria in the gums such as *Fn* may indicate periodontal status [9], and recent research has suggested that high antibody levels against these bacteria may signal increased risk for conditions such as rheumatoid arthritis and stroke, among others, while they could also impact recovery after stroke [10–12]. Moreover, other studies have indicated that *Fn* possibly has roles in promoting progression of colorectal cancer and resistance to chemotherapy [13,14]. Nevertheless, despite these associations, the clinical significance of IgG antibodies specific to *Fn* in patients with cancer has not been evaluated.

The primary goal of the present study was to clarify how pre-treatment serum levels of IgG-*Fn* are related to the clinical course and prognosis of patients with esophageal cancer. Based on the findings obtained, assessments regarding whether IgG-*Fn* could be a novel indicator for treatment response and prognosis were also performed.

## Results

### Patient characteristics

The clinical characteristics of 305 patients according to serum IgG-*Fn* status are presented in Table 1, while Table 2 shows their pathological characteristics, with those findings indicating that TNM stage differed between the groups ($p < 0.01$) Significant differences in preoperative treatment strategies were found ($p = 0.03$), with the IgG-*Fn* positive group showing a tendency to undergo surgery alone rather than with neoadjuvant chemoradiotherapy. Regarding clinical variables, no significant differences were found in regard to age, sex, tobacco smoking, tumor markers, tumor location, clinical T, N, M stages, or oral environment between the two groups.

### Survival rates for all patients based on IgG-*Fn* status

Among the 305 esophageal cancer patients enrolled, 142 died during the course of the study, with 107 deaths directly related to esophageal cancer. The average

**Table 1. Clinical characteristics of ESCC patients according to Fn IgG titer status.**

| Clinical parameters | All patients n = 305 | Serum IgG titer for *Fn* | | *p* |
| --- | --- | --- | --- | --- |
| | | Negative n = 181 | Positive n = 124 | |
| Age, mean ± SD | 65.8 ± 8.2 | 65.5 ± 8.2 | 66.2 ± 8.3 | 0.42 |
| Sex, (%) | | | | |
| Male | 249 (81.6) | 145 (80.1) | 104 (83.9) | 0.40 |
| Female | 56 (18.4) | 36 (19.9) | 20 (16.1) | |
| ECOG PS, (%) | | | | |
| 0 | 212 (69.5) | 125 (69.1) | 87 (70.2) | 0.84 |
| 1, 2 | 93 (30.5) | 56 (30.9) | 37 (29.8) | |
| Smoking history, (%) | | | | |
| Yes | 263 (86.2) | 156 (86.2) | 107 (86.3) | 0.98 |
| No | 42 (13.8) | 25 (13.8) | 17 (13.7) | |
| History of alcohol consumption, (%) | | | | |
| Yes | 272 (89.2) | 159 (87.8) | 113 (91.1) | 0.36 |
| No | 33 (10.8) | 22 (12.2) | 11 (8.9) | |
| Diabetes mellitus, (%) | | | | |
| Present | 37 (12.1) | 19 (10.5) | 18 (14.5) | 0.29 |
| Absent | 268 (87.9) | 162 (89.5) | 106 (85.5) | |
| Tumor marker (pre-treatment), mean ± SD | | | | |
| SCC, ng/mL | 2.0 ± 2.9 | 1.9 ± 2.5 | 2.3 ± 3.1 | 0.25 |
| CEA, ng/mL | 3.1 ± 2.2 | 3.1 ± 2.0 | 3.1 ± 2.4 | 0.86 |
| Tumor location, (%) | | | | |
| Upper third | 56 (18.4) | 38 (21.0) | 18 (14.5) | 0.33 |
| Middle third | 154 (50.5) | 87 (48.1) | 67 (54.0) | |
| Lower third/EGJ | 95 (31.1) | 56 (30.9) | 39 (31.5) | |
| Tumor differentiation shown by biopsy, (%) | | | | |
| Poor | 58 (19.0) | 37 (20.4) | 21 (16.9) | 0.46 |
| Other | 247 (81.0) | 144 (79.6) | 103 (83.1) | |
| Tumor depth, (%) | | | | |
| cT1/2 | 137 (44.9) | 81 (44.8) | 56 (45.2) | 0.94 |
| cT3/4 | 168 (55.1) | 100 (55.2) | 68 (54.8) | |
| Lymph node metastasis, (%) | | | | |
| cN0 | 138 (45.2) | 77 (42.5) | 61 (49.2) | 0.25 |
| cN1-3 | 167 (54.8) | 104 (57.5) | 63 (50.8) | |
| Distant metastasis, (%) (cM1: supraclavicular) | | | | |
| cM0 | 281 (92.1) | 166 (91.7) | 115 (92.7) | 0.74 |
| cM1 (LYM) | 24 (7.9) | 15 (8.3) | 9 (7.3) | |
| Clinical TNM stage, (%) | | | | |
| I | 88 (28.9) | 46 (25.4) | 42 (33.9) | 0.31 |
| II | 72 (23.6) | 48 (26.5) | 24 (19.3) | |
| III | 108 (35.4) | 64 (35.4) | 44 (35.5) | |
| IV | 37 (12.1) | 23 (12.7) | 14 (11.3) | |
| Neoadjuvant therapy, (%) | | | | |
| None | 102 (33.4) | 51 (28.2) | 51 (41.1) | 0.03* |
| Chemotherapy | 95 (31.2) | 57 (31.5) | 38 (30.7) | |
| Chemoradiotherapy | 108 (35.4) | 73 (40.3) | 35 (28.2) | |

*(Continued)*

**Table 1.** (Continued)

| Clinical parameters | All patients n = 305 | Serum IgG titer for *Fn* | | p |
|---|---|---|---|---|
| | | Negative n = 181 | Positive n = 124 | |
| Oral environment (pre-treatment), mean ± SD | (n = 213) | (n = 124) | (n = 89) | |
| Tooth loss | 8.0 ± 9.0 | 8.7 ± 9.5 | 7.2 ± 8.2 | 0.23 |
| Bleeding on probing | 32.8 ± 29.7 | 34.9 ± 32.0 | 30.0 ± 26.1 | 0.22 |

TNM staging based on the TNM Classification, 8th edition. cM1 denotes metastasis to the supraclavicular lymph node. p < 0.05 indicates statistical significance.

**Table 2.** Comparison of pathological characteristics according to serum IgG titer indicating Fn status.

| Pathological parameters | All patients n = 305 | Serum IgG titer for *Fn* | | p |
|---|---|---|---|---|
| | | Negative n = 181 | Positive n = 124 | |
| Tumor depth, (%) | | | | |
| pT0/1/2 | 205 (67.2) | 128 (70.7) | 77 (62.1) | 0.12 |
| pT3/4 | 100 (32.8) | 53 (29.3) | 47 (37.9) | |
| Lymph node metastasis, (%) | | | | |
| pN0 | 163 (53.4) | 102 (56.4) | 61 (49.2) | 0.22 |
| pN1-3 | 142 (46.6) | 79 (43.6) | 63 (50.8) | |
| Distant metastasis, (%) | | | | |
| pM0 | 291 (95.4) | 175 (96.7) | 116 (93.5) | 0.20 |
| pM (LYM) 1 | 14 (4.6) | 6 (3.3) | 8 (6.5) | |
| Pathological TNM stage, (%) | | | | |
| 0 | 35 (11.5) | 26 (14.4) | 9 (7.3) | 0.004 |
| I | 85 (27.9) | 45 (24.9) | 40 (32.2) | |
| II | 73 (23.9) | 54 (29.8) | 19 (15.3) | |
| III | 95 (31.1) | 48 (26.5) | 47 (37.9) | |
| IV | 17 (5.6) | 8 (4.4) | 9 (7.3) | |

Pathological staging based on the TNM Classification, 8th edition. pM1 (LYM) indicates pathological metastasis to supraclavicular lymph node. p < 0.05 indicates statistical significance.

duration of monitoring for survivors was 83 ± 35 months [± standard deviation (SD)]. Kaplan Meier analysis showed 3- and 5-year overall survival (OS) rates of 57.5% and 49.8%, respectively, for the IgG-*Fn*-positive patients, whereas those for the IgG-*Fn*-negative patients were higher at 76.2% and 66.6%, respectively (hazard ratio [HR] 1.59, 95% confidence interval [CI] 1.14–2.21; p = 0.01) (Fig 1A). Cancer-specific survival (CSS) after three and five years were 62.4% and 57.9%, respectively, for IgG-*Fn*-positive, and 79.5% and 72.4%, respectively, for IgG-*Fn*-negative patients (HR 1.59, 95% CI 1.09–2.33; p = 0.02) (Fig 1B). Both OS and CSS were notably reduced in the IgG-*Fn*-positive group.

Kaplan Meier survival curves for each clinical stage based on IgG-*Fn* status were also determined (Fig 2). In stage I patients, CSS rate was not significantly different based on IgG-*Fn* status (HR 0.36, 95% CI 0.07–1.81; p = 0.20) (Fig 2A). In contrast, for those in stage II, IgG-*Fn* positivity was significantly correlated with reduced survival (HR 2.94, 95% CI 1.30–6.63; p = 0.01), and this trend was shown to continue further in stage III and IV patients, with the association becoming more significant (HR 2.03, 95% CI 1.28–3.22; p = 0.002) (Fig 2B, 2C).

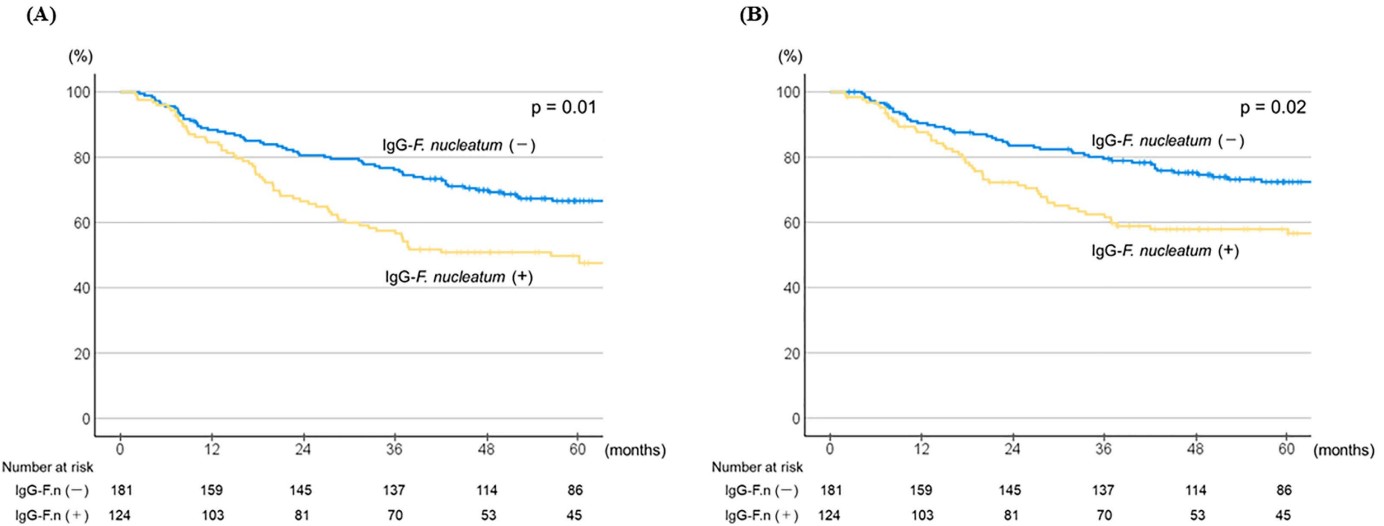

**Fig 1. Kaplan-Meier survival curves showing overall and cancer-specific survival according to serum IgG titer indicating _Fn_ status in all patients (n = 305). (A)** Overall survival for all patients (HR 1.59, 95% CI 1.14-2.21; _p_ = 0.01). **(B)** Cancer-specific survival for all patients (HR 1.59, 95% CI 1.09-2.33; _p_ = 0.02).

### Prognostic factors prior to treatment related to cancer survival in all patients

Factors associated with cancer survival in all patients were examined using univariate and multivariate analyses (Table 3). Univariate analysis showed that gender difference, along with performance status (PS), and history of smoking and drinking also played roles as prognostic indicators prior to treatment for cancer-specific survival. Results of multivariate analysis indicated that IgG-_Fn_ status was a factor for predicting outcomes for CSS (positive vs. negative: HR 1.96, 95% CI 1.32–2.90; _p_ < 0.001), PS (1/2 vs. 0: HR 1.98, 95% CI 1.31–3.00; _p_ = 0.001), and cT status (3/4 vs. 1/2: HR 2.29, 95% CI 1.10–4.76; _p_ = 0.03).

### Survival rate and prognostic factors in patients with oral environment assessment

Survival rates and prognostic factors were analyzed in 213 patients who also underwent examinations of the oral environment. Periodontal conditions before esophageal cancer treatment were evaluated in each, including number of teeth lost and bleeding on probing (BOP) (Table 1), with survival rates according to IgG-_Fn_ status and pretreatment prognostic factors assessed. For those who were IgG-_Fn_-positive, Kaplan-Meier analysis showed three- and five-year OS rates of 56.3% and 49.0%, respectively, whereas those rates for the IgG-_Fn_-negative patients were higher at 74.4% and 65.3%, respectively (HR 1.68, 95% CI 1.13–2.50; p = 0.01) (S1A Fig). CSS rates after three and five years were 61.6% and 56.5%, respectively, for the IgG-_Fn_-positive, and 78.0% and 70.8%, respectively, for the IgG-_Fn_-negative patients (HR 1.72, 95% CI 1.10–2.72; p = 0.02) (S1B Fig). Additionally, factors associated with cancer survival in this subgroup of patients were examined using univariate and multivariate analyses, which indicated that IgG-_Fn_ status was a factor for predicting CSS outcome (positive vs. negative: HR 2.22, 95% CI 1.40–3.53; p < 0.001) (S1 Table).

### Clinicopathological characteristics of patients who underwent neoadjuvant therapy (NAT)

Next, clinicopathological characteristics based on IgG-_Fn_ status of 191 patients classified from stage II to IV who received NAT were examined, with details shown in S2 Table. While no significant differences were observed in regard to the examined parameters, there were significant differences for pathological factors between the IgG-_Fn_-positive and

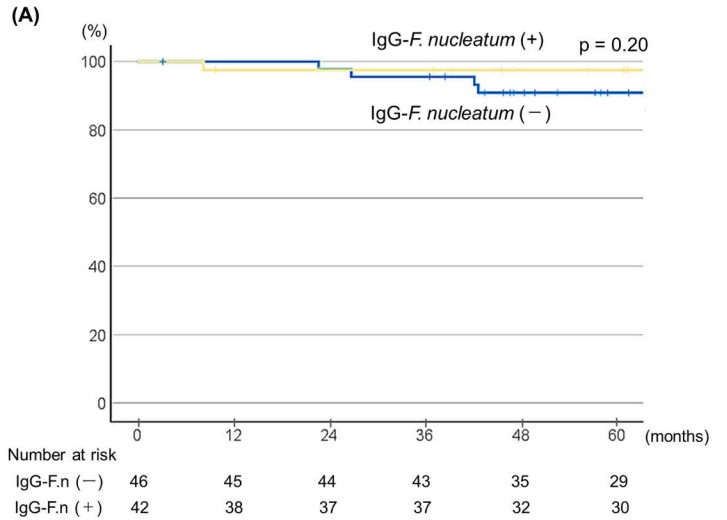

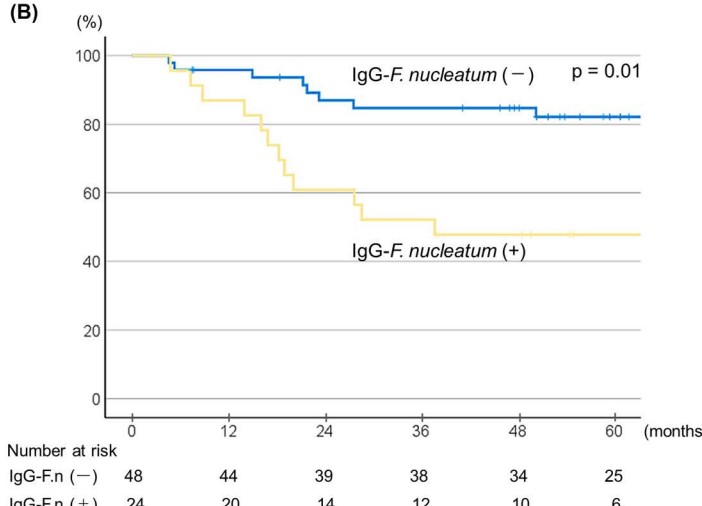

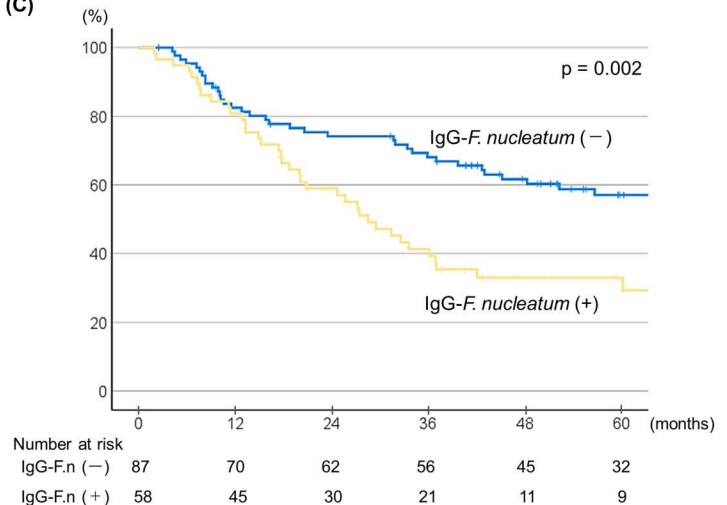

**Fig 2. Kaplan-Meier survival curves showing cancer-specific survival according to serum IgG titer indicating *Fn* status for different clinical stages (n = 305). (A)** Clinical stage I patients (HR 0.36, 95% CI 0.07-1.81; *p* = 0.20). **(B)** Clinical stage II patients (HR 2.94, 95% CI 1.30-6.63; *p* = 0.01). **(C)** Clinical stage III/IV patients (HR 2.03, 95% CI 1.28-3.22; *p* = 0.002).

-negative groups. In patients who tested positive for IgG-*Fn*, significant differences were observed for ypT (tumor depth) (0–2 vs. 3/4, *p* < 0.05), ypN (lymph node metastasis) (0 vs. 1–3, *p* = 0.02), and ypStage (TNM classification; 0-II vs. III/IV, *p* < 0.001), indicating that their condition was more advanced. Additionally, tumor regression grade (TRG) following NAT was notably worse for the IgG-*Fn*-positive as compared to the -negative group (*p* = 0.01).

## Survival rate according to serum IgG-*Fn* status in NAT group

Among patients in stage II to IV who received NAT, those in the IgG-*Fn*-positive group had an OS rate of 40.6% after three years and 33.3% after five years, and those in the IgG-*Fb*-negative group had OS rates of 72.8% and 62.7%, respectively (HR 2.28, 95% CI 1.54–3.38; *p* < 0.001) (S2A Fig). Additionally, the three- and five-year CSS rates for the IgG-*Fn*-positive group were 44.4% and 38.1%, respectively, while those for the IgG-*Fn*-negative group were higher at 75.6% and 66.9%,

**Table 3. Cox regression analysis of pretherapeutic factors for cancer-specific survival.**

| Variables | Univariate analysis | | | Multivariate analysis | | |
|---|---|---|---|---|---|---|
| | HR | 95% CI | p | HR | 95% CI | p |
| Age (continuous) | 1.02 | 0.996-1.05 | 0.09 | – | – | – |
| Female (reference: male) | 0.32 | 0.16-0.63 | <0.01* | 0.48 | 0.20-1.12 | 0.09 |
| ECOG PS 1/2 (reference: 0) | 2.20 | 1.49-3.24 | <0.001* | 1.98 | 1.31-3.00 | 0.001* |
| Body mass index (continuous) | 0.997 | 0.94-1.06 | 0.93 | – | – | – |
| Smoking history present (reference: absent) | 2.33 | 1.13-4.80 | 0.02* | 1.52 | 0.66-3.47 | 0.32 |
| Alcohol consumption present (reference: absent) | 2.53 | 1.11-5.76 | 0.03* | 1.40 | 0.54-3.62 | 0.49 |
| Diabetes mellitus present (reference: absent) | 1.20 | 0.69-2.11 | 0.52 | – | – | – |
| Upper tumor location (reference: middle, lower, or EGJ) | 1.08 | 0.66-1.77 | 0.77 | – | – | – |
| Poor differentiation in biopsy findings (reference: others) | 1.61 | 1.04-2.49 | 0.03 | 1.28 | 0.82-2.01 | 0.28 |
| CEA (pre-treatment) >5 (reference: normal value ≤5) | 1.08 | 0.60-1.93 | 0.80 | – | – | – |
| SCC (pre-treatment) >1.5 (reference: normal value ≤1.5) | 1.45 | 0.99-2.14 | 0.06 | – | – | – |
| Neoadjuvant therapy present (reference: absent) | 2.81 | 1.69-4.67 | <0.001* | 1.59 | 0.84-3.03 | 0.16 |
| cT 3/4 (reference: 1/2) | 4.00 | 2.51-6.35 | <0.001* | 2.29 | 1.10-4.76 | 0.03* |
| cN 1/2/3 (reference: 0) | 3.23 | 2.08-5.03 | <0.001* | 1.46 | 0.53-4.01 | 0.46 |
| cM (LYM) 1 (reference: 0) | 2.06 | 1.17-3.62 | 0.01* | 1.25 | 0.68-2.28 | 0.47 |
| cStage III/IV (reference: I/II) | 3.47 | 2.29-5.25 | <0.001* | 1.05 | 0.35-3.12 | 0.93 |
| Tooth loss (pre-treatment) ≥8 (reference: <8) | 1.35 | 0.88-2.07 | 0.17 | – | – | – |
| Bleeding on probing (pre-treatment) (%) ≥30 (reference: <0) | 0.95 | 0.61-1.48 | 0.81 | – | – | – |
| IgG-*Fn*-positive (reference: negative) | 1.59 | 1.09-2.33 | 0.02 | 1.96 | 1.32-2.90 | <0.001* |

Pre-therapeutic staging based on the TNM Classification, 8th edition. Clinical metastasis to the supraclavicular lymph node indicated as cM1 (LYM). p<0.05 indicates statistical significance. CI, confidence interval; HR, hazard ratio.

respectively (HR 2.33, 95% CI 1.52–3.57; p<0.001) (S2B Fig). For patients with disease stage II to IV who received NAT, both OS and CSS rates were significantly lower in the IgG-*Fn* positive group.

## Predictors of pathologic tumor response to NAT

The IgG-*Fn*-positive group exhibited more advanced pathological characteristics and worse TRG as compared to the IgG-*Fn*-negative group (S3 Table). Since IgG-*Fn* status was found to be associated with primary tumor response to NAT, analysis was performed to determine whether IgG-*Fn* could serve as a predictor. Those results revealed that patients with tumors denoted as tumor regression grade (TRG) 2 or 3 showed good response to treatment, while those with TRG 0 or 1 tumors were categorized as having poor response. Logistic regression analysis was conducted to examine factors associated with TRG 2 or 3 in contrast to TRG 0 or 1 (S3 Table). Both univariate and multivariate analysis findings revealed that undergoing chemoradiotherapy (vs. chemotherapy) [odds ratio (OR) 9.06, 95% CI: 4.62–17.80; p<0.001] as well as IgG-*Fn* level were indicative factors prior to treatment for predicting pathological response of the primary tumor (OR 0.48, 95% CI: 0.24–0.97; p=0.04).

## Discussion

Recent research has indicated that the human oral microbiome, especially periodontal pathogens, may have roles in cancer development and progression [5–7]. Understanding how these bacteria interact with cancer is important for improving cancer treatments. One such bacterium implicated in periodontal disease is *Fn*, which has been identified as a carcinogenic pathogen associated with various forms of cancer [15]. The present study retrospectively reviewed 305 ESCC patients who underwent esophagectomy surgery to determine whether IgG-*Fn* is related to prognosis. The findings

showed that IgG-*Fn*-positive patients had a worse prognosis and lower response rate to NAT, indicating that detection of IgG-*Fn* prior to starting treatment could serve as an indicator of its effects.

Several studies have investigated the relationship of *Fn* with development and advancement of tumors, in addition to examining the mechanisms of chemotherapy and radiation therapy resistance using both tumor tissues and cell lines associated with esophageal cancer. It was reported that esophageal cancer tissues exhibited high levels of *Fn* DNA as compared to normal mucosa, and a correlation between *Fn* positivity and poor prognosis was also found [7]. Additionally, *Fn* is known for its strong biofilm-forming capability, which may contribute to cancer treatment resistance. Biofilm can create a protective microenvironment for bacterial communities, shielding them from host immune responses and reducing the efficacy of chemotherapeutic agents, as well as immunotherapy and radiation therapy [16]. *Fn* invades ESCC cells, thus triggering the NF-κB pathway and increasing IL32/PRTN 3 expression, which subsequently promotes ESCC cell growth [17]. *Fn* organisms also contribute to chemotherapy and radiotherapy resistance through multiple mechanisms. One such mechanism is induction of autophagy by *Fn*, which helps cancer cells evade cisplatin-induced apoptosis. Furthermore, *Fn* enhances accumulation of myeloid-derived suppressor cells (MDSCs), which function to suppress anti-tumor immune responses [18], and has also been shown to increase m6A RNA methylation via METTL3, an epigenetic modification that promotes ESCC progression and may contribute to treatment resistance [19]. An elevated anti-*Fn* IgG level suggests prolonged exposure to *Fn* or persistence in the host rather than a simple past infection. This may activate cell survival-related pathways such as inflammatory signaling and autophagy, possibly resulting in reduced sensitivity to cytotoxic therapy.

Collectively, presented findings indicate a variety of mechanisms by which *Fn* contributes to progression of ESCC, and also chemotherapy and radiotherapy resistance. However, to the best of our knowledge, the clinical significance of serum IgG-*Fn* has not been evaluated using blood samples obtained from esophageal cancer patients. *Fn* participates in development of various malignant tumors, such as those related to colorectal, gastric, pancreatic, breast, and head and neck cancer, and promotes mechanisms related to their progression, including immune evasion, epithelial-mesenchymal transformation, and chemotherapy resistance [20,21]. In addition, a persistent presence of *Fn* in the oral cavity induces chronic inflammation, subsequently resulting in a unique microenvironment that promotes tumorigenesis [22]. These findings suggest that *Fn* plays a broader oncogenic role with complex mechanisms and effects. While the present study focused on ESCC, evaluation of serum IgG-*Fn* levels in patients at risk for *Fn*-associated malignancy may provide further insights into the utility of *Fn* as a predictive biomarker. Furthermore, those with an elevated serum IgG-*Fn* level could have increased risk of developing *Fn*-related cancer and may require long-term surveillance. The present results are the first to show a significant association of serum IgG-*Fn* with sensitivity to NAT and the prognosis of esophageal cancer patients.

Findings obtained in randomized controlled trials have shown benefits of administering NAT to patients with ESCC for enhancing long-term survival rate, which has led to the standardization of preoperative chemotherapy and chemoradiotherapy for advanced ESCC [22,23]. Nevertheless, despite the increased use of these treatments, the efficacy of NAT remains suboptimal, as approximately half of such treated patients exhibit response, while only a minority, around 20–30%, achieve complete pathological remission [24]. Therefore, it is considered important to elucidate factors that can identify patients who will respond positively to NAT. Molecular biomarkers, including mRNAs and miRNAs, have been found to predict response to NAT in ESCC patients [25–28]. Additionally, a model that incorporates liquid biopsy assay findings was found to demonstrate a high level of accuracy for predicting NAT response [29].

IgG-*Fn* was analyzed in the present study and positive patients were found to be more likely to have an unfavorable outcome following NAT about advanced stage ESCC in those who underwent surgery. This suggests that patients with a poor prognosis cannot be explained by selection bias of early-stage cases alone, as many in the high IgG-*Fn* group had not received preoperative treatment. It is thus considered that IgG-*Fn* can serve as a useful biomarker for predicting how patients with ESCC will respond to NAT and determining an appropriate treatment strategy. There is currently no standard method to predict NAT reactivity, therefore, testing related to IgG-*Fn* may help improve treatment selection. Furthermore, use of multiple markers in combination can improve predictability and provide important guidelines regarding treatment

decisions for high risk ESCC patients. Notably, the prognostic impact of IgG-*Fn* was found to vary according to stage. In Stage I patients, the survival rate was high and number of events was low, thus limiting the ability to detect significance. In contrast, in advanced clinical stages, tumor-associated inflammation and immunosuppression were more pronounced. Thus, sustained *Fn* exposure in such cases may promote immune dysregulation and chemotherapy resistance, leading to enhancement of the prognostic significance of IgG-*Fn*.

Previous studies have shown that serum IgG antibody titers related to specific periodontal pathogens are linked to periodontal condition [9,30]. Another report presented findings of serum antibody titers for *Porphyromonas gingivalis* (*Pg*) and *Fn* determined in both periodontitis patients and healthy subjects, which showed that IgG-*Pg* levels were clearly higher in periodontitis patients, whereas the IgG-*Fn* levels were not significantly different between those groups [31]. The present results showing no significant difference in oral condition according to IgG-*Fn* positivity confirm those findings, thus it is considered that IgG-*Fn* status can indicate systemic circumstances as well as oral condition. Oral care performed by dental doctors is routinely provided by our institution following hospital admission for surgery or neoadjuvant therapy. However, while oral care may result in a reduced *Fn* level over time, changes in serum antibody titers are generally more gradual. Therefore, the pre-treatment IgG-*Fn* levels noted in the present study are considered likely to reflect long-term bacterial exposure rather than short-term reduction due to oral intervention. The relationship of oral care with serum IgG-*Fn* level remains unclear, and it will be important for future studies to investigate whether such sustained care has an influence and also the prognostic significance. Additionally, the relationship between serum and tissue *Fn* status should be evaluated to understand their concordance. A previous study of colorectal cancer cases examined the relationship of serum *Fn* antibodies with tissue *Fn* DNA status and found that the rate of positive for *Fn* antibodies in patients with a high amount of *Fn* in tissue was significantly greater than in those negative for *Fn*, while the levels of *Fn* IgA and IgG antibodies in colorectal cancer patients were higher as compared to healthy controls [32]. Additional in-depth investigations will be necessary to explore correlations in ESCC patients among oral health condition, *Fn* status in the tumor tissue and IgG-*Fn* status in ESCC patients.

The present study has some limitations. First, it was conducted in a retrospective manner at a single institution with a relatively small number of patients, thus larger studies will be necessary to provide more robust findings. Furthermore, there may have been bias about patient selection due to variations in preoperative treatment strategies based on the clinical situation. Additionally, while the findings indicate an association of IgG-*Fn* positivity with NAT resistance as well as prognosis in ESCC cases, they do not provide direct experimental or functional validation to establish a causal relationship, and further investigations will be required to elucidate the precise biological mechanisms. Finally, antibiotic use prior to serum sample collection, systemic inflammation, and patient comorbidities were factors not considered in this study, though each may have an influence IgG-*Fn* status. Nevertheless, it might be valuable to examine use of pretreatment serum *Fn* status for prediction of tumor response and survival, even when considering potential confounders. Moreover, the feasibility of IgG-*Fn* testing as well as its cost effectiveness remain uncertain. While the present ELISA-based assay was a cost-efficient method for research, standardized protocols and cost-effective testing methods will be necessary for widespread clinical applications.

In conclusion, IgG-*Fn* was found to be associated with poor prognosis, suggesting its potential role as a prognostic biomarker of ESCC in patients who have undergone an esophagectomy. Additionally, the results suggest that routine examinations conducted to determine serum IgG-*Fn* status prior to treatment may contribute to development of more effective treatment strategies, particularly for patients with advanced ESCC.

## Methods

### Patients and sample collection procedures

In the present study, findings for 305 patients who had pathological condition diagnosed as ESCC based on biopsy results were retrospectively reviewed before undergoing treatment or an esophagectomy at Hiroshima University Hospital between January 2010 and December 2020. Exclusion criteria were as follows: 1) no radical resection (pathological R1

or R2), 2) unavailability of serum sample, 3) salvage esophagectomy following definitive chemoradiotherapy, and 4) total pharyngolaryngoesophagectomy for cervical esophageal cancer. All data used were obtained from clinical databases and electronic medical records. Tumor staging was determined according to the Union for International Cancer Control (8th edition) [33]. The study procedures were approved by the Ethical Committee of Hiroshima University (approval no. E-1874) and written informed consent was obtained from each of the enrolled patients. All methods were performed in accordance with the relevant guidelines and regulations. Clinical data were accessed for research purposes between 01/02/2023 and 30/04/2023. The authors had access to identifiable patient information, but all data were anonymized before analysis.

### Treatment strategy

At our institution, treatment strategies are principally performed as shown following. Patients with cT1N0 are treated by surgery alone, with adjuvant chemotherapy administered when pathological lymphatic metastasis has been established. Those with a more advanced clinical stage and who refuse NAT or definitive chemoradiation therapy are treated with surgery alone when a radical resection procedure is considered feasible. Our institution performs NAT for patients with locally advanced ESCC according to treatment strategy rationale previously described [34–36].

Among the present cohort, NAC was performed for patients with results indicating cT1-3, N-positive, or cT2N0 with two cycles of cisplatin, 5-fluorouracil (CF), or docetaxel plus CF(DCF). Neoadjuvant chemoradiotherapy (NCRT) was performed for patients with results indicating cT2-3, along with dysphagia or a bulky tumor, along with concurrent radiotherapy (40 Gy in 20 fractions) and two cycles of CF or DCF. For patients with results indicating cT4, discussion regarding their condition was conducted in a conference that included expert radiologists, surgeons, and oncologists to determine which treatment was better and decide the strategy. In patients with poor renal function who received either NAC or NCRT, cisplatin in the CF regimen was changed to nedaplatin. Surgery was performed within three to eight weeks following the final NAT treatment day when a radical resection was considered feasible.

### Pathological primary tumor regression grading criteria

Histopathologic primary tumor response to NAT was classified into four categories according to criteria noted in the Japanese Classification of Esophageal Cancer [37], as follows: grade 0 (no recognizable cytologic or histological therapeutic effect), grade 1 (slightly effective, with viable cancer cells accounting for one-third to two-thirds of the tumor tissue), grade 2 (moderately effective, with viable cancer cells accounting for less than one-third of tumor tissue, and grade 3 (markedly effective, with no viable cancer cells evident and pathological complete response). Subsequently, patients with a grade 2 or 3 tumor were classified as showing response, while those with a grade 0 or 1 tumor were classified as a poor responder.

### Determination of serum IgG-*Fn*

Peripheral blood was collected prior to treatment initiation, including neoadjuvant therapy, then centrifuged at 3000 rpm for 10 minutes at 4°C, after which the supernatant was collected and stored at −80°C until use. Serum IgG-*Fn* level was determined using an enzyme-linked immunosorbent assay, as previously described in detail [11]. Briefly, sonicated preparations of *Fn ATCC 10953* (*subspecies: polymorphum*) were used as the bacterial antigen for the present study. *ATCC 10953* is the standard strain with an oral origin, and has been used in previous studies and shown to have high comparability [38,39]. It was chosen as the representative antigen for the present study because of its common antigenicity among the four subspecies. Serum samples from five healthy subjects confirmed to not have a medical history of cancer or periodontal disease were also obtained, then pooled and used for calibration. Bacterial antigen-coated wells were washed with phosphate-buffered saline with Tween 20 (PBST), then serum samples diluted 1:3200 in PBST were added to the wells. Serial dilutions (1:800–1:12,800) of representative serum samples were prepared to confirm that assay

values were proportional to the dilution factor, with a standard dilution of 1:3200 considered to be the most stable condition and thus employed. Following incubation at 4°C overnight, the wells were washed with PBST, then filled with alkaline phosphatase-conjugated goat anti-human IgG (gamma-chain specific; Abcam, Cambridge, MA, USA) in PBST. After another incubation at 37°C for two hours, the wells were again washed with PBST, then an aliquot of p-nitrophenyl phosphate at 1 mg/ml (Wako Pure Chemical Industries Ltd., Osaka, Japan) in 10% diethanolamine buffer was added to each well as a substrate and incubation was performed at 37°C until color development was visually apparent. Optical density at 405 nm was determined using a microplate reader (iMark; Bio-Rad Laboratories Inc., Hercules, CA, USA).

The cut-off points for reactivity (positive decision point) to each antigen were defined as greater than the mean EU + 3 SD obtained from the control subjects. While previous studies have commonly used a cut-off point of 2 SD above the mean to define seropositivity [11,12], the more stringent threshold of 3 SD was used in this study to reduce false positive findings and ensure higher specificity.

## Oral assessment and care

Since April 2012, patients scheduled to undergo a procedure for esophageal cancer, including surgery, chemotherapy, or radiotherapy, are referred to the Department of Dentistry at Hiroshima University Hospital before treatment. All such patients, except those who state their refusal, receive dental care during the perioperative period to prevent complications such as oral mucositis associated with chemotherapy and/or postoperative complications. For these cases, experienced dentists perform the usual oral examinations, including determination of number of teeth lost and bleeding on probing (BOP), a parameter for monitoring periodontal conditions in clinical practice [40]. BOP is determined using fully erupted teeth at six sites per tooth with a pressure-sensitive probe, with bleeding noted as either present or absent within 30 seconds of probing, as previously described [41]. In the present study, 213 of the 305 patients received such evaluations of oral conditions before undergoing treatment for esophageal cancer (Table 1).

## Statistical analysis

Mean values were compared using Student's t-test for continuous variables and a χ2 test for categorical variables. OS was defined as time from date of surgery to date of death due to any cause or the most recent follow-up examination. CSS was defined as the period from date of surgery to date of death attributable to ESCC or the most recent follow-up examination. Survival outcomes were evaluated in March 2024 using Kaplan-Meier curves, with differences between groups assessed using a log-rank test (two-sided).

A $p$-value <0.05 obtained in univariate analysis was considered to indicate a clinically meaningful factor, which was then further analyzed using multivariate Cox proportional hazards regression and logistic regression analyses to identify independent predictors. For these analyses, normal cutoff values for the pretherapeutic tumor markers carcinoembryonic antigen (CEA) and squamous cell carcinoma-related antigen (SCC) were used (5.0 and 1.5, respectively). Cutoff values for tooth loss and BOP were determined based on the approximate mean values (8 and 30, respectively). Additionally, sensitivity analysis of the 213 cases in which oral indicators were obtained was conducted. All statistical tests were performed as two-sided analyses with a significance level of 0.05, using SPSS version 27 (IBM Corporation, Armonk, NY, USA), with actual $P$ values presented where applicable.

## Supporting information

**S1 Fig. Kaplan-Meier survival curves showing overall and cancer-specific survival in patients who underwent assessment of the oral environment according to serum IgG titer indicating Fn status (n = 213).** (A) Overall survival (HR 1.68, 95% CI 1.13–2.50; p = 0.01). (B) Cancer-specific survival (HR 1.72, 95% CI 1.10–2.72; p = 0.02). (TIF)

**S2 Fig. Kaplan-Meier survival curves for clinical stage II-IV patients who underwent NAT followed by surgery, according to serum IgG titer indicating *Fn* status (n = 191).** (A) Overall survival (HR 2.28, 95% CI:1.54–3.38; $p < 0.001$). (B) Cancer-specific survival (HR 2.33, 95% CI 1.52–3.57; $p < 0.001$).
(TIF)

**S1 Table. Cox regression analysis of pretherapeutic factors for cancer-specific survival in patients who underwent oral environment assessment (n = 213).** Univariate and multivariate Cox proportional hazards models for cancer-specific survival. Covariates include pre-treatment clinical factors and serum IgG-Fn status. HR > 1 indicates increased hazard of cancer-specific death.
(DOCX)

**S2 Table. Clinicopathological characteristics of ESCC patients undergoing neoadjuvant therapy by IgG-Fn titer (n = 191).** Clinicopathological characteristics of ESCC patients receiving neoadjuvant therapy based on IgG-Fn titer status, with TNM staging based on the TNM Classification, 8th edition. ypStage 0 and ypT0 denotes complete pathological response, with ypStage 0 showing no lymph node metastasis. pM1(LYM) denotes metastasis to the supraclavicular lymph node. p < 0.05 indicates significance.
(DOCX)

**S3 Table. Logistic regression analysis of pre-therapeutic factors for pathological tumor response to neoadjuvant therapy (TRG 2–3 vs 0–1) in stage II–IV ESCC patients (n = 191).** Logistic regression analyses of pretherapeutic factors for pathological primary tumor response (Grade 2/3), based on the response evaluation criteria of the Japan Esophageal Society. p < 0.05 indicates statistical significance. CI, confidence interval; OR, Odds ratio.
(DOCX)

## Acknowledgments

We thank all the staff of Hiroshima University Hospital, who provided excellent care for the patients in the present study.

## Author contributions

**Conceptualization:** Toru Yoshikawa, Hiromi Nishi, Manabu Emi, Morihito Okada.

**Data curation:** Toru Yoshikawa, Yoichi Hamai, Yuta Ibuki, Tomoaki Kurokawa, Ryosuke Hirohata, Manato Ohsawa, Nao Kitasaki.

**Formal analysis:** Toru Yoshikawa, Hiromi Nishi, Manabu Emi, Yoichi Hamai.

**Project administration:** Hiromi Nishi.

**Supervision:** Hitoshi Komatsuzawa, Hiroyuki Kawaguchi, Morihito Okada.

**Writing – original draft:** Toru Yoshikawa.

**Writing – review & editing:** Hiromi Nishi, Manabu Emi, Yoichi Hamai, Morihito Okada.

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
