## [Decision Letter · Decision Letter 0]

18 Sep 2025

Dear Dr. Nishi,

Thank you for submitting your manuscript to PLOS ONE. After careful consideration, we feel that it has merit but does not fully meet PLOS ONE’s publication criteria as it currently stands. Therefore, we invite you to submit a revised version of the manuscript that addresses the points raised during the review process. Specifically, the reviewer recommends revisions that would improve statistical reporting and methodological rigor.

We look forward to receiving your revised manuscript.

Kind regards,

David M. Ojcius

Academic Editor

PLOS ONE

4. We notice that your supplementary table is included in the manuscript file. Please remove them and upload them with the file type 'Supporting Information'. Please ensure that each Supporting Information file has a legend listed in the manuscript after the references list.

Reviewers' comments:

Reviewer's Responses to Questions

**Comments to the Author**

1. Is the manuscript technically sound, and do the data support the conclusions?

Reviewer #1: Yes

2. Has the statistical analysis been performed appropriately and rigorously?

Reviewer #1: Yes

3. Have the authors made all data underlying the findings in their manuscript fully available?

Reviewer #1: Yes

4. Is the manuscript presented in an intelligible fashion and written in standard English?

Reviewer #1: Yes

Reviewer #1: This retrospective cohort study explores the prognostic utility of pre-treatment serum IgG antibody titers against Fusobacterium nucleatum (IgG-Fn) in 305 esophageal squamous cell carcinoma patients undergoing esophagectomy. The central finding—that IgG-Fn positivity associates with reduced survival and diminished response to neoadjuvant therapy—addresses a clinically relevant gap in ESCC biomarker research. While the cohort size is appropriate and statistical methods generally sound, several revisions are necessary to strengthen methodological rigor and contextual interpretation.

Specific comments

1.The ELISA methodology requires significant elaboration to ensure reproducibility. Specifically, authors should detail the rationale for selecting the F. nucleatum subspecies beyond citing ATCC 10953, explain how serum dilution ratios were optimized. Clarity regarding sample timing is essential—explicitly confirm all specimens were obtained before any treatment initiation, including neoadjuvant therapy. The partial oral health assessment in just 213 patients introduces possible selection bias that warrants explanation or sensitivity analysis.

2.Statistical reporting needs refinement throughout. All Kaplan-Meier figures must be supplemented with "Number at Risk" tables and annotated with hazard ratios to improve interpretability. Critical inconsistencies between text sections require resolution, such as the abstract's misstatement about IgG-Fn-negative patients showing worse pathology—this directly contradicts Supplementary Table 1 findings where IgG-Fn-positive status correlates with adverse pathological features.

3.Interpretation would benefit from deeper biological contextualization. The stark contrast in IgG-Fn's prognostic significance between Stage I versus advanced-stage patients necessitates mechanistic exploration—is this attributable to statistical power limitations, fundamental biological differences in early disease, or the modifying effect of neoadjuvant therapy exposure? While chemotherapy resistance mechanisms are cited, the discussion fails to integrate how the IgG biomarker itself relates to these pathways—does elevated IgG indicate failed bacterial clearance that enables Fn-mediated treatment resistance?

**Do you want your identity to be public for this peer review?** For information about this choice, including consent withdrawal, please see our Privacy Policy

Reviewer #1: No

---

## [Author Response · Author response to Decision Letter 1]

17 Oct 2025

Dear Editor,

Thank you for providing us with an opportunity to resubmit our study. Please find our detailed responses to each comment below.

Comment 1

The ELISA methodology requires significant elaboration to ensure reproducibility. Specifically, authors should detail the rationale for selecting the F. nucleatum subspecies beyond citing ATCC 10953, explain how serum dilution ratios were optimized. Clarity regarding sample timing is essential—explicitly confirm all specimens were obtained before any treatment initiation, including neoadjuvant therapy. The partial oral health assessment in just 213 patients introduces possible selection bias that warrants explanation or sensitivity analysis.

Authors’ response:

Thank you for these valuable suggestions. The rationale for selecting ATCC 10953 F. nucleatum subsp. polymorphum) among the four subspecies of Fusobacterium nucleatum for ELISA and steps taken to adopt a 1:3200 dilution is now described in the revised Methods section (lines 366-369, 373-375). In addition, to clarify the timing of blood collection, it is noted that it was performed before the start of any treatment in all cases (Methods, lines 361-362).

Based on these excellent comments and suggestions, results of sensitivity analysis regarding the response to partial oral health assessment have been added to the revised manuscript. Similar to the main analysis, the findings obtained with the subset of 213 patients showed that IgG-Fn positivity was consistently associated with poor prognosis (OS: HR=1.68, 95% CI 1.13-2.50; p=0.01) (CSS: HR=1.72, CI 1.10-2.72; p=0.02) (multivariate Cox CSS: HR=2.22, CI 1.40-3.53; p<0.001). These results have been added to the Methods (lines 417-418) and Results (lines 139-152) sections and are also presented in Supplementary Figure 1 and Supplementary Table 1.

Comment 2

Statistical reporting needs refinement throughout. All Kaplan-Meier figures must be supplemented with "Number at Risk" tables and annotated with hazard ratios to improve interpretability. Critical inconsistencies between text sections require resolution, such as the abstract's misstatement about IgG-Fn-negative patients showing worse pathology—this directly contradicts Supplementary Table 1 findings where IgG-Fn-positive status correlates with adverse pathological features.

Authors’ response:

We appreciate these helpful comments. We have added number at risk indicating results of Kaplan-Meier analyses to all the figures and also added hazard ratios to the figure legends (lines 102-133). In addition, the misstatement in the original Abstract has been corrected (line 34).

Comment 3

Interpretation would benefit from deeper biological contextualization. The stark contrast in IgG-Fn's prognostic significance between Stage I versus advanced-stage patients necessitates mechanistic exploration—is this attributable to statistical power limitations, fundamental biological differences in early disease, or the modifying effect of neoadjuvant therapy exposure? While chemotherapy resistance mechanisms are cited, the discussion fails to integrate how the IgG biomarker itself relates to these pathways—does elevated IgG indicate failed bacterial clearance that enables Fn-mediated treatment resistance?　

Authors’ response:

Thank you for these insightful comments. The following has been added to the Discussion section in the revised version.

Notably, the prognostic impact of IgG-Fn was found to vary according to stage. In Stage I patients, the survival rate was high and number of events was low, thus limiting the ability to detect a significant effect. In contrast, in advanced clinical stages, tumor-associated inflammation and immunosuppression were more pronounced. Thus, sustained Fn exposure in such cases may promote immune dysregulation and chemotherapy resistance, leading to enhancement of the prognostic significance of IgG-Fn. (lines 292-298).

An elevated anti-Fn IgG level suggests prolonged exposure to Fn or persistence in the host rather than a simple past infection. This may activate cell survival-related pathways such as inflammatory signaling and autophagy, possibly resulting in reduced sensitivity to cytotoxic therapy. (lines 251-254).

---

## [Decision Letter · Decision Letter 1]

23 Oct 2025

Serum IgG titer findings for Fusobacterium nucleatum associated with clinical outcome following surgery in patients with esophageal squamous cell carcinoma

PONE-D-25-26990R1

Dear Dr. Nishi,

We’re pleased to inform you that your manuscript has been judged scientifically suitable for publication and will be formally accepted for publication once it meets all outstanding technical requirements.

Kind regards,

David M. Ojcius

Academic Editor

PLOS ONE

Reviewers' comments:

Reviewer's Responses to Questions

**Comments to the Author**

Reviewer #1: All comments have been addressed

2. Is the manuscript technically sound, and do the data support the conclusions?

Reviewer #1: Yes

3. Has the statistical analysis been performed appropriately and rigorously?

Reviewer #1: Yes

4. Have the authors made all data underlying the findings in their manuscript fully available?

Reviewer #1: Yes

5. Is the manuscript presented in an intelligible fashion and written in standard English?

Reviewer #1: Yes

Reviewer #1: The authors have addressed all my concerns, I think this manuscript is suitable for publication in PLOS One. Thank you very much.

**Do you want your identity to be public for this peer review?** For information about this choice, including consent withdrawal, please see our Privacy Policy

Reviewer #1: No

---

## [Editor Report · Acceptance letter]

PONE-D-25-26990R1

PLOS ONE

Dear Dr. Nishi,

I'm pleased to inform you that your manuscript has been deemed suitable for publication in PLOS ONE. Congratulations! Your manuscript is now being handed over to our production team.

Kind regards,

on behalf of

Dr. David M. Ojcius

Academic Editor

PLOS ONE